# Pairing instabilities of the Yukawa-SYK models with controlled fermion incoherence

**Wonjune Choi[1,2], Omid Tavakol[3] and Yong Baek Kim[3⋆]**

**1** Department of Physics, Technical University of Munich, 85748 Garching, Germany
**2** Munich Center for Quantum Science and Technology (MCQST),
Schellingstr. 4, 80799 München, Germany
**3** Department of Physics, University of Toronto, Toronto, Ontario M5S 1A7, Canada

⋆ ybkim@physics.utoronto.ca

## Abstract

The interplay of non-Fermi liquid and superconductivity born out of strong dynamical interactions is at the heart of the physics of unconventional superconductivity. As a solvable platform of the strongly correlated superconductors, we study the pairing instabilities of the Yukawa-Sachdev-Ye-Kitaev (Yukawa-SYK) model, which describes spin-1/2 fermions coupled to bosons by the random, all-to-all, spin-independent and dependent Yukawa interactions. In contrast to the previously studied models, the random Yukawa couplings are sampled from a collection of Gaussian ensembles whose variances follow a continuous distribution rather than being fixed to a constant. By tuning the analytic behaviour of the distribution, we could control the fermion incoherence to systematically examine various normal states ranging from the Fermi liquid to non-Fermi liquids that are different from the conformal solution of the SYK model with a constant variance. Using the linearised Eliashberg theory, we show that the onset of the unconventional spin-triplet pairing is preferred with the spin-dependent interactions while all pairing channels show instabilities with the spin-independent interactions. Although the interactions shorten the lifetime of the fermions in the non-Fermi liquid, the same interactions also dress the bosons to strengthen the tendency to pair the incoherent fermions. As a consequence, the onset temperature $T_c$ of the pairing is enhanced in the non-Fermi liquid compared to the case of the Fermi liquid.



# 1  Introduction

Understanding unconventional superconductivity of strongly correlated electrons is a long-standing goal of modern condensed matter physics [1–10]. It is generally believed that dynamical interactions mediated by collective charge or spin fluctuations are responsible for the Cooper pairing in the correlated superconductors [11,12]. Major challenges of the problem come from the emergence of non-Fermi liquid normal states due to the same dynamical interactions [13–18]. Since both superconductivity and non-Fermi liquid are stabilized by the same physical origin, systematic investigations of two competing effects are necessary [19–22]. However, the strongly coupled nature of the problem makes it difficult to draw a concrete theoretical conclusion as no small parameter exists to control the theory perturbatively.

In this work, we circumvent such difficulty by examining a variant of the Sachdev-Ye-Kitaev (SYK) model [23–27], so-called the Yukawa-SYK model [28–31], which is exactly solvable and supports non-Fermi liquid ground states. While the previously studied models use the fixed variance of the random coupling, we introduce a continuous distribution of variances. The model consists of $N$ number of fermions ($c_{i=1,\dots,N}$) strongly coupled to $M$ number of bosons ($\phi_{k=1,\dots,M}$) via the random all-to-all Yukawa couplings ($g_{ij,k}$):

$$H_{\text{int}} = \sum_{i,j=1}^{N} \sum_{k=1}^{M} g_{ij,k} c_{i\alpha}^{\dagger} \sigma_{\alpha\beta}^{a} c_{j\beta} \phi_k \,, \tag{1}$$

where $\sigma^a$ is the Pauli matrix acting on the spin space $\alpha, \beta = \uparrow, \downarrow$, and the summation is assumed for the repeated Greek indices. Physically, the scalar bosons $\phi_k$ represent the collective charge (or spin) fluctuations of the fermion bilinear $\sum_{i,j} c_{i\alpha}^{\dagger} c_{j\alpha}$ $\left(\text{or } \sum_{i,j} c_{i\alpha}^{\dagger} \sigma_{\alpha\beta}^{3} c_{j\beta}\right)$. The recurring theme of the SYK model and its variants is that the disorder averaging over the random coupling constants ($g_{ij,k}$) suppresses almost all quantum fluctuations except one or a few families of the Feynmann diagrams in the large $M$ and $N$ limits [23,24,32]. With those handful number of diagrams, we can solve the model self-consistently and identify the leading pairing instabilities without any perturbative approximations.

It is important to note that the Yukawa-SYK model is defined by not only the Hamiltonian but also the statistical properties of the random couplings ($g_{ij,k}$). Most of previous studies on various families of the SYK model focused on the random couplings with zero mean ($\overline{g_{ij,k}} = 0$) and constant variance ($\overline{(g_{ij,k})^2} = \lambda$) [26–31,31,33–38]. However, we can also consider the random couplings whose variances obey a well-defined distribution, i.e., $\overline{(g_{ij,k})^2} = \lambda_k$ has the $k$ dependence such that the set of the variances $\{\lambda_k\}$ forms a continuous distribution $\rho(\lambda)$ in the large $M$ limit. Pioneering work on the low-rank SYK models [39], which are equivalent

to the Yukawa-SYK models with the extensive ($M/N \sim \mathcal{O}(1)$) number of nondynamical massive bosons, first notices the significance of the distribution $\rho(\lambda)$; depending on the singular behaviour of the distribution $\rho(\lambda)$ near the maximum variance $\lambda_{\mathrm{max}}$, the low-rank SYK models show a rich variety of the low energy states ranging from the Fermi liquid to non-Fermi liquids [39, 40]. By tuning the distribution $\rho(\lambda)$, we can systematically control the fermion incoherence and push the system toward the non-Fermi liquid. Therefore the current variant of the Yukawa-SYK model is an excellent solvable platform to examine the interplay of non-Fermi liquid and superconductivity with the distribution $\rho(\lambda)$ as a theoretical handle to control the incoherence of fermions.

While the flourishing papers discussed the SYK superconductivity, they focused on the fast scrambling conformal solution of the SYK model (and its variants) with a fixed constant variance [28, 29, 34–38]. The pairing instabilities of the Fermi liquid and the nonconformal non-Fermi liquid states of the low-rank SYK models are not examined yet [39]. Since the variance distribution $\rho(\lambda)$ opens up a new direction of the controllability for the "non-Fermi-liquidness", we would like to understand whether the strong interaction, which makes the fermions more incoherent but the bosons to glue the fermions stronger, is an ally or a foe of the Cooper pair formation. The enhanced transition temperatures $T_c$ of the non-Fermi liquid state (Figure 3) demonstrate that the highly incoherent fermions can prefer the pairing more than the well-defined quasiparticles of the Fermi liquid due to the significant enhancement of the bosonic glue in the Yukawa-SYK model. Furthermore, to understand the distinct contributions of the collective charge and spin fluctuations to the pairing, we examine both the spin-singlet and triplet pairing instabilities with the linearized Schwinger-Dyson equations. The unconventional dynamical pairing between the equal-spin fermions at distinct times, i.e., $\langle c_\uparrow^\dagger(\tau) c_\uparrow^\dagger(0) \pm c_\downarrow^\dagger(\tau) c_\downarrow^\dagger(0) \rangle \neq 0$, is found to occur.

The remaining part of the paper is organized as follows. In Sec. 2, we introduce the Yukawa-SYK model and its effective action in terms of the Green functions and self-energies. Sec. 3 discusses the Schwinger-Dyson equations, which are the saddle point equations of the effective action. We first consider the high-temperature normal state solutions in Sec. 4, which demonstrate how the distribution of variances can result in both the Fermi liquid and non-Fermi liquids. Then, in Sec. 5, we discuss the leading pairing instabilities of the Fermi liquid and the non-Fermi liquid normal states by solving the linearized Schwinger-Dyson equations. At last, we summarize and conclude our work in Sec. 6.

## 2 Model

We consider spin-1/2 fermions ($c$) coupled to real scalar fields ($\phi$) by all-to-all random Yukawa couplings ($g$), $S = S_c + S_\phi + S_g$:

$$S_c = \int_0^\beta d\tau \sum_{i=1}^N c_{i\alpha}^\dagger \frac{d}{d\tau} c_{i\alpha}, \tag{2}$$

$$S_\phi = \frac{1}{2} \int_0^\beta d\tau \sum_{k=1}^M \phi_k \left( -\frac{d^2}{d\tau^2} + m^2 \right) \phi_k, \tag{3}$$

$$S_g = \frac{1}{N} \int_0^\beta d\tau \sum_{i,j=1}^N \sum_{k=1}^M g_{ij,k} c_{i\alpha}^\dagger \sigma_{\alpha\beta}^a c_{j\beta} \phi_k. \tag{4}$$

We use the natural unit $\hbar = k_{\mathrm{B}} = 1$ so that $\beta = 1/T$ is the inverse temperature. Since $S_c$ and $S_\phi$ are invariant under spin rotation, it is sufficient to investigate two cases: $a = 0$ and $a = 3$.

The real symmetric Yukawa couplings $g_{ij,k} = g_{ji,k} \in \mathbb{R}$ are sampled from the Gaussian

orthogonal ensemble (GOE) for each $k$, i.e., $g_{ij,k}$ follows the Gaussian distribution with zero mean $\overline{g_{ij,k}} = 0$ and variance $\overline{g_{ij,k}g_{i'j',k'}} = \lambda_k \delta_{k,k'}(\delta_{ii'}\delta_{jj'} + \delta_{ij'}\delta_{ji'})$ for $\lambda_k > 0$. Assuming that the model is self-averaging, we can derive the effective action from the disorder average of the partition function $Z$:

$$e^{-S_\lambda} = \overline{e^{-S_g}} = \exp\left[\sum_{ij,k}\frac{\lambda_k}{4N^2}\left(A_{ij,k} + A^\dagger_{ij,k}\right)^2\right], \tag{5}$$

where $A_{ij,k} = \int_0^\beta d\tau\, c^\dagger_{i\alpha}\sigma^a_{\alpha\beta}c_{j\beta}\phi_k$ (see Appendix A for the derivation). Note that the pairing correlations among fermions $(A_{ij,k})^2 \sim (c^\dagger_{i\alpha}c^\dagger_{i\alpha'})(c_{j\beta}c_{j\beta'})$ are generated because the random Yukawa couplings are averaged over GOE [36].

With the bilocal fields

$$D(\tau,\tau') = \frac{1}{M}\sum_{k=1}^{M}\lambda_k\phi_k(\tau')\phi_k(\tau), \tag{6}$$

$$G_{\alpha\alpha'}(\tau,\tau') = \frac{1}{N}\sum_{i=1}^{N}c^\dagger_{i\alpha'}(\tau')c_{i\alpha}(\tau), \tag{7}$$

$$F_{\alpha\alpha'}(\tau,\tau') = \frac{1}{N}\sum_{i=1}^{N}c_{i\alpha'}(\tau')c_{i\alpha}(\tau), \tag{8}$$

$$F^+_{\alpha\alpha'}(\tau,\tau') = \frac{1}{N}\sum_{i=1}^{N}c^\dagger_{i\alpha'}(\tau')c^\dagger_{i\alpha}(\tau), \tag{9}$$

we can rewrite the interacting part of the effective action $S_\lambda$ defined in Eq. (5):

$$S_\lambda = \frac{\gamma N}{2}\int_0^\beta d\tau\, d\tau'\, D(\tau',\tau)\Big[G_{\sigma'\sigma}(\tau',\tau)\sigma^a_{\sigma\rho}G_{\rho\rho'}(\tau,\tau')\sigma^a_{\rho'\sigma'}$$
$$-F^+_{\sigma'\sigma}(\tau',\tau)\sigma^a_{\sigma\rho}F_{\rho\rho'}(\tau,\tau')(\sigma^a)^T_{\rho'\sigma'}\Big], \tag{10}$$

where $\gamma = M/N \sim \mathcal{O}(1)$ is the ratio between the number of bosons and fermions. Note that $D(\tau,\tau')$ is the bilocal field that becomes the sum of the bosonic propagators weighted by the variances $\lambda_k$ at the saddle point of the action. By introducing the Lagrange multipliers $\Sigma$, $\Phi^+$, $\Phi$, and $\Pi$, we can relate the dynamics of the fermions and bosons with the bilocal fields $G$, $F$, $F^+$, and $D$, respectively (see Appendix A). Physically, the bilocal fields become the fermionic ($G, F, F^+$) and bosonic ($D$) Green functions, and the Lagrange multipliers become the corresponding fermion ($\Sigma, \Phi^+, \Phi$) and boson ($\Pi$) self-energies, at the saddle point of the action.

In this model, the bosonic part of the action $\widetilde{S}_\phi = S_\phi + S_\Pi$ (see Appendix A for the definition of the bosonic self-energy action $S_\Pi$) needs special attention because the bosons may condense at low temperatures. After the Fourier transformations, we split $\widetilde{S}_\phi$ into the normal [Eq. (11)] and condensed parts [Eq. (12)]:

$$\widetilde{S}_\phi = \sum_{k=1}^{M}\sum_{n=1}^{\infty}\left(\nu_n^2 + m^2 - \lambda_k\Pi(i\nu_n)\right)|\phi_k(i\nu_n)|^2 \tag{11}$$

$$+ \frac{1}{2}\sum_{k=1}^{M}\left(m^2 - \lambda_k\Pi(0)\right)(\phi_k(0))^2, \tag{12}$$

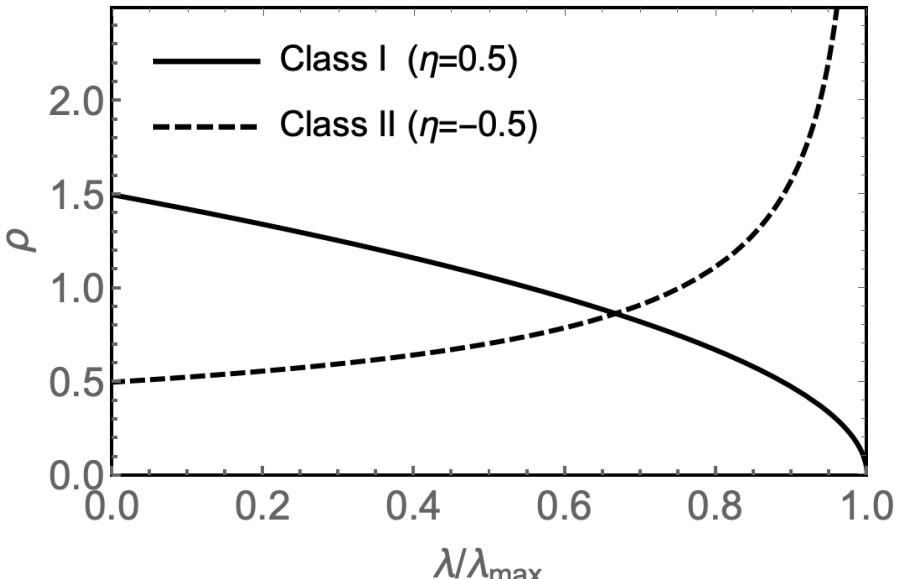

Figure 1: Model distributions of the variances $\rho_\eta(\lambda)$ for the Yukawa-SYK model. Depending on the sign of $\eta$, the class I ($\eta > 0$) and class II ($\eta < 0$) distributions show qualitatively different behaviour at $\lambda = \lambda_{\max}$

where $\nu_n = 2\pi n/\beta$ are the bosonic Matsubara frequencies. The bosons are condensed when the quadratic potential for some bosonic modes is no longer convex. As the zero frequency modes $\phi_{\bar{k}}(0)$ with $\lambda_{\bar{k}} = \lambda_{\max} = \max[\{\lambda_k\}]$ first become unstable when $m^2 - \lambda_{\max}\Pi(0) = 0$, they are condensed at $T < T_{\text{BEC}}$ [39]. Then

$$\varphi = \frac{1}{\beta N} \sum_{k:\lambda_k = \lambda_{\max}} (\phi_k(0))^2 \tag{13}$$

can be treated as a classical degree of freedom. By integrating out the fermions and remaining uncondensed bosons, we can obtain the large $N$ effective action $S_{\text{eff}}$ in terms of the bilocal fields and the Lagrange multiplier fields (see Appendix A).

## 3 Schwinger-Dyson Equations

In the large $M$ and $N$ limits, the saddle point of $S_{\text{eff}}$ precisely describes the low-energy dynamics of the Yukawa-SYK model. Hence, we derive the Schwinger-Dyson equations from the condition $\delta S_{\text{eff}} = 0$.

The normal ($G$) and anomalous ($F$) Green functions for the fermions are

$$G(i\omega_n) = \left[i\omega_n\sigma^0 - \Sigma(i\omega_n) - \Phi(i\omega_n)\left[i\omega_n\sigma^0 + \Sigma(-i\omega_n)^T\right]^{-1}\Phi^+(i\omega_n)\right]^{-1}, \tag{14}$$

$$F(i\omega_n) = G(i\omega_n)\Phi(i\omega_n)\left[i\omega_n\sigma^0 + \Sigma(-i\omega_n)^T\right]^{-1}, \tag{15}$$

where the spin indices are suppressed for notational convenience.

We assume that the set of variances $\{\lambda_k\}$ forms a well-defined distribution $\rho(\lambda)$ in the large $M$ limit:

$$\rho_\eta(\lambda) = \frac{1}{M}\sum_{k=1}^{M}\delta(\lambda - \lambda_k) = \frac{1+\eta}{\lambda_{\max}^{1+\eta}}(\lambda_{\max} - \lambda)^\eta, \tag{16}$$

which is regular at $\lambda = \lambda_{\max}$ for $\eta > 0$ (class I) but diverges algebraically as $\lambda \to \lambda_{\max}$ for $-1 < \eta < 0$ (class II) (Figure 1) [39]. Then the bosonic propagator is

$$
\begin{aligned}
D(i\nu_n) &= \frac{\beta}{\gamma}\lambda_{\max}\varphi\,\delta_{n,0} + \int_0^{\lambda_{\max}} \frac{\lambda\rho_\eta(\lambda)d\lambda}{\nu_n^2 + m^2 - \lambda\Pi(i\nu_n)} \\
&\equiv \frac{\beta}{\gamma}\lambda_{\max}\varphi\,\delta_{n,0} + D_N(i\nu_n).
\end{aligned}
\tag{17}
$$

The first part of Eq. (17) comes from the condensed bosons, and the latter part $D_N(i\nu_n)$ is from the uncondensed bosons. $\varphi \neq 0$ if $m^2 - \lambda_{\max}\Pi(0) = 0$, and $\varphi = 0$ otherwise. The low-energy properties of $D_N(i\nu_n)$ depend on the analytic behaviour of $\rho_\eta(\lambda)$ near $\lambda_{\max}$ because the bosonic modes with $\lambda \sim \lambda_{\max}$ have light effective mass $m^2 - \lambda\Pi(i\nu_n)$ at small frequencies $\nu_n$.

With $M, N \to \infty$, the self-energies for the fermions and bosons satisfy the Schwinger-Dyson equations:

$$
\Sigma(i\omega_n) = \frac{\gamma}{\beta}\sum_{m\in\mathbb{Z}} D(i\nu_m)\sigma^a G(i\nu_m + i\omega_n)\sigma^a,
\tag{18}
$$

$$
\Phi(i\omega_n) = -\frac{\gamma}{\beta}\sum_{m\in\mathbb{Z}} D(i\nu_m)\sigma^a F(i\nu_m + i\omega_n)(\sigma^a)^T,
\tag{19}
$$

$$
\begin{aligned}
\Pi(i\nu_n) = &-\frac{1}{\beta}\sum_{m\in\mathbb{Z}} \mathrm{tr}\big[G(i\omega_m)\sigma^a G(i\omega_m + i\nu_n)\sigma^a\big] \\
&- \mathrm{tr}\big[F^+(i\omega_m)\sigma^a F(i\omega_m + i\nu_n)(\sigma^a)^T\big],
\end{aligned}
\tag{20}
$$

where the spin indices for $G$, $F$, $\Sigma$, and $\Phi$ are suppressed for simpler notation, and "tr" is the trace over the spin indices. After we plug in Eqs. (14) and (15) to Eqs. (18 – 20), we can get a set of nonlinear equations for the normal and anomalous fermionic self-energies $\Sigma(i\omega_n)$ and $\Phi(i\omega_n)$. Since the fermions would not be paired at high temperatures, our analysis starts from the normal state with $F(i\omega_n) = \Phi(i\omega_n) = 0$.

## 4 Normal State Analysis

Without the pairing among fermions, Eq. (14) gives the fermion Green function

$$
G_\alpha(i\omega_n) \equiv G_{\alpha\alpha}(i\omega_n) = \frac{1}{i\omega_n - \Sigma_\alpha(i\omega_n)},
\tag{21}
$$

where $\Sigma_\alpha(i\omega_n) \equiv \Sigma_{\alpha\alpha}(i\omega_n)$. For both $a = 0, 3$ in $S_g$, the fermion Green function is spin-diagonal ($G_{\uparrow\downarrow} = G_{\downarrow\uparrow} = 0$) and independent of spin polarization ($G_{\uparrow\uparrow} = G_{\downarrow\downarrow}$). Therefore we write $G_0(i\omega_n) \equiv G_\uparrow(i\omega_n) = G_\downarrow(i\omega_n)$ and $\Sigma_0(i\omega_n) \equiv \Sigma_\uparrow(i\omega_n) = \Sigma_\downarrow(i\omega_n)$, where

$$
\begin{aligned}
\Sigma_0(i\omega_n) &= \frac{\gamma}{\beta}\sum_{n'\in\mathbb{Z}} D(i\nu_{n'})G_0(i\nu_{n'} + i\omega_n) \\
&= \lambda_{\max}\left(\varphi + \frac{\gamma}{\beta\lambda_{\max}}\int_0^{\lambda_{\max}}\frac{\lambda\rho_\eta(\lambda)d\lambda}{m^2 - \lambda\Pi(0)}\right)G_0(i\omega_n) \\
&\quad + \frac{\gamma}{\beta}\sum_{n'\neq 0}\int_0^{\lambda_{\max}}\frac{\lambda\rho_\eta(\lambda)d\lambda}{\nu_{n'}^2 + m^2 - \lambda\Pi(i\nu_{n'})}G_0(i\nu_{n'} + i\omega_n) \\
&\equiv \lambda_{\max}\tilde{\varphi}\,G_0(i\omega_n) + \frac{\gamma}{\beta}\sum_{n'\neq 0} D_N(i\nu_{n'})G_0(i\nu_{n'} + i\omega_n) \tag{22} \\
&\equiv \Sigma_C(i\omega_n) + \Sigma_N(i\omega_n), \tag{23}
\end{aligned}
$$

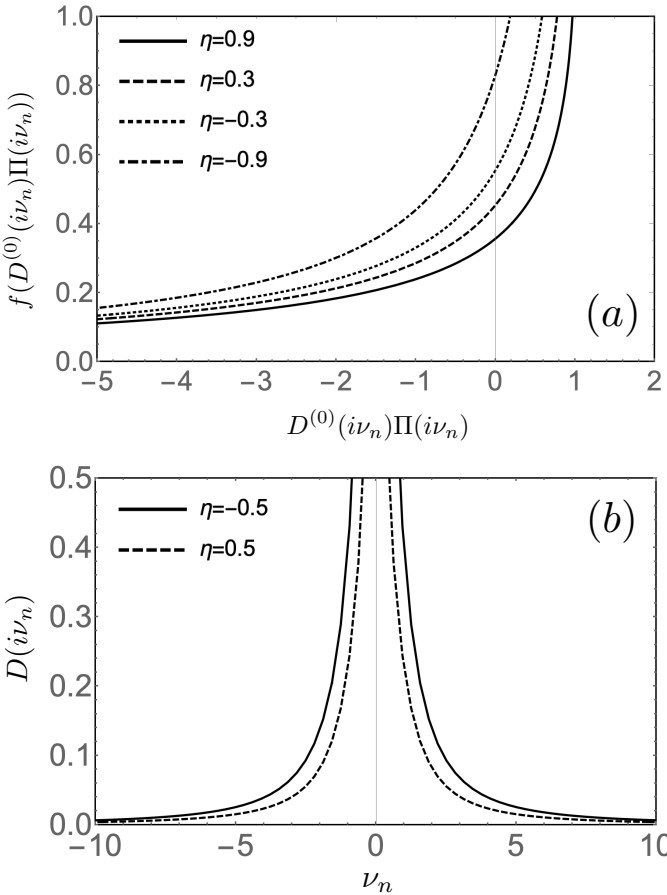

Figure 2: The propagator for uncondensed bosons for $\gamma = \lambda_{\max} = m = 1$, $D(i\nu_n) = D^{(0)}(i\nu_n)f_\eta(D^{(0)}(i\nu_n)\Pi(i\nu_n))$. (a) The positive, monotonic function $f_\eta$ has larger value for smaller $\eta$, i.e., $f_\eta(x) < f_{\eta'}(x)$ if $\eta > \eta'$. (b) The class II bosonic propagator ($\eta < 0$) is larger than the class I propagator ($\eta > 0$) for all frequency range. Since the distribution $\rho_{\eta<0}(\lambda)$ is mostly concentrated around $\lambda \sim \lambda_{\max}$, there is high chance to sample strong Yukawa coupling $g_{ij,k}$. Hence, the bosonic propagator is more strongly enhanced by the interactions between fermions and bosons.

and the effective condensate $\tilde\varphi = \varphi + \gamma D_N(0)/\beta\lambda_{\max}$. Then the fermion Green function

$$iG_0(i\omega_n) = \frac{2}{J(i\omega_n) + \operatorname{sgn}(J(i\omega_n))\sqrt{J(i\omega_n)^2 + 4\lambda_{\max}\tilde\varphi}} \tag{24}$$

solves the Schwinger-Dyson equation with $J(i\omega_n) = \omega_n + i\Sigma_N(i\omega_n)$ [39].

With our model distribution $\rho_\eta(\lambda)$ in Eq. (16), the propagator for the uncondensed bosons $D_N(i\nu_n)$ is

$$D_N(i\nu_n) = \frac{\lambda_{\max}}{\nu_n^2 + m^2}\int_0^{\lambda_{\max}} \frac{d\lambda}{\lambda_{\max}}\frac{\lambda\rho_\eta(\lambda)}{1 - \lambda\Pi(i\nu_n)/(\nu_n^2 + m^2)} \tag{25}$$

$$= D^{(0)}(i\nu_n)f_\eta(D^{(0)}(i\nu_n)\Pi(i\nu_n)), \tag{26}$$

where $D^{(0)}(i\nu_n) = \lambda_{\max}/(\nu_n^2 + m^2)$, and

$$f_\eta(x) = \frac{2 + \eta - (1+\eta)_2F_1(1,1;3+\eta;x)}{(2+\eta)(1-x)}. \tag{27}$$

The function $f_\eta$ is positive and monotonic, and $f_\eta(x) < f_{\eta'}(x)$ for a given $x$ if $\eta > \eta'$ [Figure 2 (a)]. Note that the distribution $\rho_\eta(\lambda)$ shows larger value near $\lambda = \lambda_{\max}$ when $\eta$ is smaller, i.e., $\rho_\eta(\lambda \sim \lambda_{\max}) < \rho_{\eta'}(\lambda \sim \lambda_{\max})$ when $\eta > \eta'$. Since $D(i\nu_n)$ is the bosonic propagator weighted by the variance $\lambda_k$ [Eq. (6)], it is enhanced when there is higher chance to sample the Yukawa couplings $g_{ij,k}$ with large variance $\lambda_k$.

The asymptotic expansion of the hypergeometric function ${}_2F_1(a, b; c; x)$ gives

$$f_\eta(x) = \frac{1}{\eta} + \frac{\pi(1+\eta)}{\sin\pi(1+\eta)}(1-x)^\eta + \dots \tag{28}$$

$$\sim \begin{cases} 1/\eta + c_\eta(1-x)^\eta, & \eta > 0, \\ c_\eta(1-x)^\eta, & -1 < \eta < 0, \end{cases} \tag{29}$$

near $x = 1$ with $c_\eta = \pi(1+\eta)/\sin\pi(1+\eta)$. By self-consistently solving the Schwinger-Dyson equations, we can check that the bosonic self-energy $\Pi(i\nu_n)$ is a decreasing function of positive $\nu_n$. Thus, $\nu_n \sim 0$ implies $x \sim 1$. So the asymptotic expansion well approximates the low-frequency behaviour of $D(i\nu_n)$.

A qualitatively important distinction between class I ($\eta > 0$) and class II ($\eta < 0$) is the boundedness of the Green function for uncondensed bosons, $D_N(i\nu_n)$. While $f_\eta(x) \leq 1/\eta$ is bounded from above for class I ($\eta > 0$), $f_\eta(x)$ diverges algebraically as $x \to 1^-$ for class II ($-1 < \eta < 0$). Thus, the boson Green function $D(i\nu_n)$ for the class I model is bounded from above if there were no Bose-Einstein condensation. When the boson Green function is bounded from above, the bosons can yield limited quantum corrections to the fermion self-energy. Hence, the fermion self-energy also becomes bounded from above. Without large fermion self-energy, the strong Yukawa interactions result in large boson self-energy which can make the bosons unstable, i.e., the renormalized squared mass of the bosons $m_{\text{ren}}^2 = m^2 - \lambda\Pi(0)$ can be negative due to the large boson self-energy at zero frequency $\Pi(0)$. Hence, the zero frequency bosons need to be condensed when $x = \lambda_{\max}\Pi(0)/m^2 = 1$. If the bosons are condensed, the total boson Green function $D(i\nu_n) = (\beta/\gamma)\lambda_{\max}\varphi\delta_{n,0} + D_N(i\nu)$ is no longer bounded from above because the Bose condensate $\varphi$ is not bounded from above. Therefore, the Bose-Einstein condensation is essential for the class I model ($\eta > 0$). On the other hand, the class II model ($\eta < 0$) or the model with a fixed variance coupling $\rho(\lambda) \sim \delta(\lambda - \lambda_0)$ do not show the Bose-Einstein condensation. More detailed mathematical discussion about the Bose-Einstein condensation in the Yukawa-SYK model can be found in Appendix B.

In the absence of the pairing $F = \Phi = 0$, the same Schwinger-Dyson equations are solved in the context of the low-rank SYK models, which can be obtained from the Yukawa-SYK models by integrating out the massive bosons. Since the asymptotic expansion of our bosonic propagator, Eq. (29), coincides with the bosonic propagator in Ref. [39], thermodynamics of the Yukawa-SYK models are equal to that of the low-rank SYK models. For the self-contained discussion, we will briefly review the frequency scaling and thermodynamics of the normal state Green functions and self-energies [39].

The distribution of variances $\rho_\eta(\lambda)$ plays important role in the frequency scaling of the fermion Green function and the self-energy. For small frequencies at low temperatures,

$$G(i\omega_n) \sim i\,\text{sgn}(\omega_n), \tag{30}$$

$$\Sigma_N(i\omega_n) \sim i\,\text{sgn}(\omega_n)|\omega_n|^{1+\eta}. \tag{31}$$

The self-consistency of the above frequency scaling properties can be checked as follows. Since the overall argument for class I ($\eta > 0$) and class II ($-1 < \eta < 0$) are similar, let us focus on class II because the frequency scaling of the self-energy will play important role in non-Fermi liquid behaviours. If Eq. (31) is true, then the fermion self-energy $\Sigma_N(i\omega_n) \to 0$ as $\omega_n \to 0$ for $\eta > -1$. Hence, from Eq. (24), $G_0(i\omega_n) \sim i\,\text{sgn}(\omega_n)/\sqrt{\lambda_{\max}\bar{\varphi}} \sim i\,\text{sgn}(\omega_n)$ for small

frequencies at low temperatures. This implies $G_0(\tau) \sim 1/\tau$ after the Fourier transformation. Then the boson self-energy $\Pi(\tau) \sim G_0(\tau)G_0(-\tau) \sim 1/|\tau|^2$. Hence,

$$\Pi(\omega) - \Pi(0) = 2 \int_0^\infty \Pi(\tau)(\cos \omega\tau - 1)\, d\tau \sim |\omega|.$$

From the asymptotic expansion in Eq. (29), the Green function for the uncondensed bosons

$$D_N(\omega) \sim \left(1 - \frac{\lambda_{\max}\Pi(\omega)}{m^2}\right)^\eta \sim (\Pi(0) - \Pi(\omega))^\eta \sim |\omega|^\eta,$$

for $-1 < \eta < 0$ in the low frequency limit $\omega \ll m$. Also, we used the numerical observation that $\Pi(0) \approx m^2/\lambda_{\max}$ at low temperatures. Since

$$\Sigma_N(\tau) \sim D_N(\tau)G_0(\tau) \sim \left(\frac{1}{\tau^{1+\eta}}\right)\left(\frac{1}{\tau}\right) = \frac{1}{\tau^{2+\eta}},$$

$\Sigma_N(\omega) \sim i\,\mathrm{sgn}(\omega)|\omega|^{1+\eta}$. Therefore, the frequency scalings in Eqs. (30) and (31) are self-consistent. More detailed numerical and analytical investigations about the scaling properties can be found in Ref. [39].

At low temperatures, the low-frequency dynamics of bosons are important to determine the thermodynamics of the normal state. By approximating the free energy as a saddle point action, i.e., $\beta F = S_{\mathrm{eff}}$ such that $\delta S_{\mathrm{eff}} = 0$, one can show that the leading contribution to the energy density comes from the effective condensate $\tilde{\varphi}$, i.e., $E/N \approx -\tilde{\varphi}/2$ [39]. The effective condensate $\tilde{\varphi}$ can be approximately calculated from the condition

$$\frac{m^2}{\lambda_{\max}} \approx \Pi(0) = \frac{2}{\beta} \sum_{n\in\mathbb{Z}} [iG_0(i\omega_n)]^2$$

$$= \frac{1}{\beta} \sum_{n=0}^\infty \frac{16}{\left[J(i\omega_n) + \sqrt{J(i\omega_n)^2 + 4\lambda_{\max}\tilde{\varphi}}\right]^2} \equiv \frac{1}{\beta} \sum_{n=0}^\infty g(\tilde{\varphi}, i\omega_n) \tag{32}$$

$$= \int_{\pi/\beta}^\infty g(\tilde{\varphi}, \omega) \frac{d\omega}{2\pi} + \frac{1}{2\beta} g(\tilde{\varphi}, \pi/\beta) - \frac{\pi}{6\beta^2} \frac{\partial g}{\partial \omega}(\tilde{\varphi}, \pi/\beta) + \ldots \tag{33}$$

$$= \int_0^\infty g(\tilde{\varphi}, \omega) \frac{d\omega}{2\pi} - \int_0^{\pi/\beta} \left[g(\tilde{\varphi}, \pi/\beta) + \frac{\partial g}{\partial \omega}(\tilde{\varphi}, \pi/\beta)(\omega - \pi/\beta) + \ldots\right] \frac{d\omega}{2\pi}$$

$$\qquad\qquad + \frac{1}{2\beta} g(\tilde{\varphi}, \pi/\beta) - \frac{\pi}{6\beta^2} \frac{\partial g}{\partial \omega}(\tilde{\varphi}, \pi/\beta) + \ldots \tag{34}$$

$$= \int_0^\infty g(\tilde{\varphi}, \omega) \frac{d\omega}{2\pi} + \frac{\pi}{12\beta^2} \frac{\partial g}{\partial \omega}(\tilde{\varphi}, \pi/\beta) + \ldots. \tag{35}$$

In the third line, the Euler-McLaurin formula was used to approximate the series. We can solve the integral equation with the successive approximation. Let $\tilde{\varphi} = \tilde{\varphi}_0 + \delta\tilde{\varphi}$ such that $\frac{m^2}{\lambda_{\max}} - \int_0^\infty g(\tilde{\varphi}_0, \omega) \frac{d\omega}{2\pi} = 0$, i.e., $\tilde{\varphi}_0$ is the effective condensate at zero temperature. At low temperatures, $\delta\tilde{\varphi}$ must be small. So we expand Eq. (35) with respect to $\delta\tilde{\varphi}$:

$$\frac{m^2}{\lambda_{\max}} - \int_0^\infty g(\tilde{\varphi}, \omega) \frac{d\omega}{2\pi} = -\left[\int_0^\infty g_1(\tilde{\varphi}_0, \omega) \frac{d\omega}{2\pi}\right]\delta\tilde{\varphi} \tag{36}$$

$$= \frac{\pi}{12\beta^2} \frac{\partial g}{\partial \omega}(\tilde{\varphi}, \pi/\beta) + \ldots$$

$$= \left[\frac{2\pi}{3}(4\lambda_{\max}\tilde{\varphi}_0)^{-3/2}\left(1 + \frac{1}{\sqrt{4\lambda_{\max}\tilde{\varphi}_0}}\right)\right]\frac{1}{\beta^2}\frac{dJ}{d\omega}(\pi/\beta) + \ldots, \tag{37}$$

where $g(\check{\varphi}, \omega) = g(\check{\varphi}_0, \omega) + g_1(\check{\varphi}_0, \omega)\delta\check{\varphi} + \mathcal{O}\left((\delta\check{\varphi})^2\right)$. Then, one can easily read out the temperature dependence of $\delta\check{\varphi} \sim J'(\pi/\beta)/\beta^2$.

Although the fermion self-energy $|i\Sigma_N(i\omega_n)| \sim |\omega_n|^{1+\eta} \ll \omega_n$ is negligible for class I ($\eta > 0$) at low frequencies, $|i\Sigma_N(i\omega_n)| \sim |\omega_n|^{1+\eta} \gg \omega_n$ is significant in class II ($-1 < \eta < 0$). Hence, in the low frequency limit ($|\omega_n| \to 0$),

$$
J(i\omega_n) \sim \begin{cases} \omega_n, & \eta > 0, \\ \mathrm{sgn}(\omega_n)|\omega_n|^{1+\eta}, & -1 < \eta < 0, \end{cases} \tag{38}
$$

which implies

$$
E/N \approx -\frac{1}{2}\check{\varphi} \sim \frac{1}{\beta^2}J'(\pi/\beta) \sim \begin{cases} T^2, & \eta > 0, \\ T^{2+\eta}, & -1 < \eta < 0. \end{cases} \tag{39}
$$

Thus, the heat capacity

$$
C_V \sim \begin{cases} T, & \eta > 0, \\ T^{1+\eta}, & -1 < \eta < 0, \end{cases} \tag{40}
$$

demonstrates the Fermi-liquid-like behaviour for class I and a non-Fermi liquid property of the class II Yukawa-SYK model [39]. While the class I ($\eta > 0$) shows conventional linear temperature dependence, the class II ($-1 < \eta < 0$) exhibits anomalously large heat capacity at low temperatures because of algebraically diverging $\rho(\lambda) \to \infty$ as $\lambda \to \lambda_{\max}$.

Note that these two new classes of the normal states are qualitatively different from the quantum critical SYK non-Fermi liquid (SYK-NFL) normal state of the Yukawa-SYK model with a fixed variance coupling, $\rho(\lambda) \sim \delta(\lambda - \lambda_0)$ [28, 36, 41]. The SYK-NFL state is the fast scrambling conformal solution of the fixed variance model, and its non-Fermi liquid nature originates from the strong boson-fermion interactions which dynamically tune the mass of the bosons to zero [28]. The scaling property of the fermion Green function $G(i\omega_n)$ for the SYK-NFL state is dominated by the fermion self-energy $\Sigma(i\omega_n) \sim i\mathrm{sgn}(\omega_n)|\omega_n|^{1-2\Delta}$ at low frequencies, and the scaling dimension, which lies between $\frac{1}{4} < \Delta < \frac{1}{2}$, depends on the ratio between the number of bosons ($M$) and fermions ($N$), $\gamma = M/N$.

On the contrary, the normal states of our model with the distribution of the variances $\rho(\lambda)$ show different scaling behaviour. Both "Fermi liquid" ($\eta > 0$) and "non-Fermi liquid" ($-1 < \eta < 0$) states show the "impurity-like" behaviour, which has been observed from the fixed-variance model at intermediate temperature window [36, 41]. The strong interactions do not tune the mass of the boson to zero. Instead, the bosons act as impurity centres that scatter fermions. Especially, as we can see from Eq. (24), the impurity-like scattering due to the Bose condensate $\varphi$ and the static uncondensed bosons $D_N(i\nu_0 = 0)$ lead to the leading scaling dimension of the fermions $\Delta = \frac{1}{2}$, i.e., $G(i\omega_n) \sim i\mathrm{sgn}(\omega_n)$, for all $\gamma$ and $\eta > -1$. While the impurity-like non-Fermi liquid fixed point is not stable in the fixed-variance models, our model supports the impurity-like Fermi liquid and non-Fermi liquid states as stable infrared fixed points at low temperatures.

## 5 Pairing Instabilities of Fermi and Non-Fermi Liquids

We are interested in pairing instabilities of fermions in the presence of the singlet ($a = 0$) and the triplet ($a = 3$) Yuakawa interactions [Eq. (4)]. Hence, we consider not only singlet pairing but also triplet pairings. Let us expand the anomalous part of the Green function and

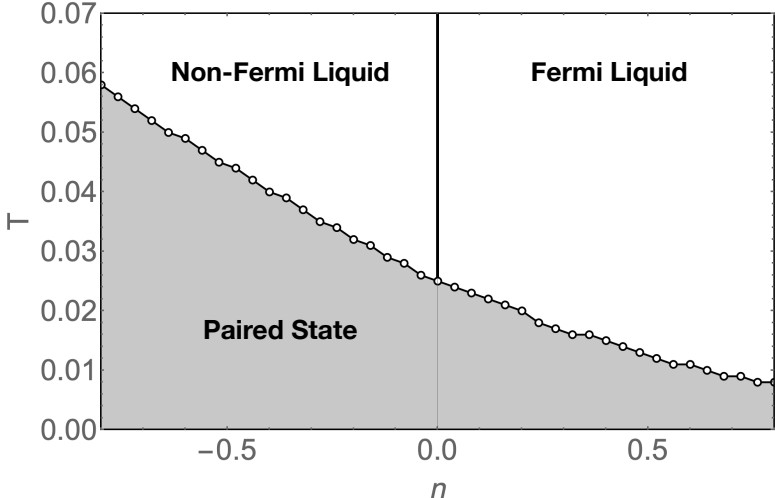

Figure 3: Phase diagram of the Yukawa-SYK model with $\gamma = \lambda_{\max} = m = 1$. The phase boundary demonstrates the leading pairing instabilities of the normal state. The singlet Yukawa coupling has the instabilities in all pairing channels at the same temperatures. However, the triplet Yukawa coupling shows the pairing instabilities only in the spin-preserving triplet channels. Both the singlet and triplet couplings have the same transition temperature $T_c$ for a given $\eta$ that determines the distribution of the variances $\rho_\eta(\lambda)$.

the self-energy in the singlet ($\mu = 0$) and the triplet channels ($\mu = 1, 2, 3$):

$$F(i\omega_n) = \sum_{\mu=0}^{3} F^\mu(i\omega_n) i\sigma^2\sigma^\mu, \tag{41}$$

$$\Phi(i\omega_n) = \sum_{\mu=0}^{3} \Phi^\mu(i\omega_n) i\sigma^2\sigma^\mu. \tag{42}$$

Then Eq. (19) becomes

$$\Phi^\mu(i\omega_n) = -\frac{\gamma}{\beta} \sum_{m\in\mathbb{Z}} \zeta D(i\nu_m) F^\mu(i\nu_m + i\omega_n), \tag{43}$$

where $\zeta = 1$ if $\sigma^a$ and $\sigma^2\sigma^\mu$ commutes and $\zeta = -1$ if $\sigma^a$ and $\sigma^2\sigma^\mu$ anticommutes. Hence, $\zeta = 1$ for all pairing channels ($\mu = 0, 1, 2, 3$) in case of the singlet Yukawa coupling ($a = 0$). However, $\zeta = 1$ for $\mu = 1, 2$ and $\zeta = -1$ for $\mu = 0, 3$ in case of the triplet Yukawa coupling ($a = 3$).

At the critical temperatures $T_c$, we consider a continuous phase transition to a paired state. Near $T_c$, the anomalous part of the self-energy $\Phi(i\omega_n)$ and the Green function $F(i\omega_n)$ must be very small. Hence, we linearize the Schwinger-Dyson equations to estimate $T_c$ and identify the leading pairing instability. Then we can approximate the anomalous Green function $F(i\omega_n)$ with the normal state Green function $G_0(i\omega_n)$ near $T_c$:

$$F^\mu(i\omega_n) = -G_0(i\omega_n)\Phi^\mu(i\omega_n)G_0(-i\omega_n) \tag{44}$$

$$= -(iG_0(i\omega_n))^2\Phi^\mu(i\omega_n) = -\frac{\Phi^\mu(i\omega_n)}{(\omega_n + i\Sigma_0(i\omega_n))^2}. \tag{45}$$

In the second line, we used the odd parity of $G_0(i\omega_n) = -G_0(-i\omega_n)$. Then we get the linearized Schwinger-Dyson equations for the paring channels:

$$\Phi^\mu(i\omega_n) = \frac{\zeta}{\beta} \sum_{m\in\mathbb{Z}} \frac{\gamma D(i\omega_m - i\omega_n)}{(\omega_m + i\Sigma_0(i\omega_m))^2} \Phi^\mu(i\omega_m). \tag{46}$$

Since the bosonic propagator is in the numerator while the fermionic self-energy is in the denominator of Eq. (46), strong Yukawa couplings lead to two competing effects: enhancement of the bosonic propagator $D$, which is the pairing glue of fermions, and decoherence of fermions due to large fermionic self-energy $\Sigma_0$.

Using the bosonic propagator and fermionic self-energy of the normal state, we can calculate the transition temperature $T_c$ from the condition that the linearized equation, Eq. (46), has the nontrivial solution. Figure 3 shows the phase diagram of the Yukawa-SYK model for the various distribution parameter $\eta$. The phase boundary implies the leading pairing instabilities of the model. While the non-Fermi liquid states with $\eta < 0$ (class II) are known to have large fermionic self-energy $\Sigma_N(i\omega_n) \sim |\omega_n|^{1+\eta}$ (compared to the free Green function $G_{\text{free}}(i\omega_n)^{-1} \sim \omega_n$) due to the uncondensed bosons [39], their transition temperatures are greater than those of the Fermi liquid states with $\eta > 0$. Our result implies that the enhancement of the pairing glue $D(i\nu_n)$ (Figure 2) plays a more important role in the pairing than the decoherence of fermions in the Yukawa-SYK model.

While the singlet coupling ($a = 0$) yields the same linearized equations for both singlet ($\mu = 0$) and triplet pairing channels ($\mu = 1, 2, 3$), the triplet Yukawa coupling ($a = 3$) turns out to have the attractive pairing channels only for the spin-preserving triplet pairing ($\mu = 1, 2$). Note that the spin-preserving triplet pairings are

$$F^1(\tau) = \sum_{j=1}^N \langle c_{j\uparrow}^\dagger(\tau)c_{j\uparrow}^\dagger(0) - c_{j\downarrow}^\dagger(\tau)c_{j\downarrow}^\dagger(0) \rangle, \tag{47}$$

$$F^2(\tau) = \sum_{j=1}^N i\langle c_{j\uparrow}^\dagger(\tau)c_{j\uparrow}^\dagger(0) + c_{j\downarrow}^\dagger(\tau)c_{j\downarrow}^\dagger(0) \rangle. \tag{48}$$

Due to the Pauli exclusion principle, these pairings must be vanishing in the static limit $\tau \to 0$. Only the dynamical pairing among fermions at distinct times can be finite. Therefore, the leading pairing instabilities with the triplet Yukawa coupling ($a = 3$) correspond to the dynamical pairing of fermions. Such a feature is distinguished from the conventional pairing in the BCS theory. Apart from the nature of the paired states, the transition temperature $T_c$ for both the singlet and triplet Yukawa-SYK models are the same for a given value of $\eta$. Hence, the phase diagrams for the singlet and triplet couplings are the same although the paired states' nature is different.

# 6 Conclusion

In summary, we present a solvable strongly coupled theory of spin-half fermions $c_{i\sigma}$ interacting with scalar bosons $\phi_k$ by the all-to-all random Yukawa couplings $g_{ij,k}$. For each boson $\phi_k$, the Yukawa coupling constant $g_{ij,k}$ is sampled from the Gaussian orthogonal ensemble of zero mean, $\overline{g_{ij,k}} = 0$, and finite variance, $\overline{(g_{ij,k})^2} = \lambda_k$. With a large number of fermions and bosons, we assume that the theory is self-averaging and the set of the variances $\{\lambda_k\}$ forms a continuous distribution $\rho(\lambda)$ (Figure 1). An important aspect of the theory is the systematic controllability of the fermionic incoherence with the distribution $\rho(\lambda)$ responsible for the statistical nature of the Yukawa interaction $g_{ij,k}$. The model can realize both the Fermi liquid normal state

when $\rho(\lambda)$ is regular at the maximum variance $\lambda_{\text{max}}$ and the non-Fermi liquid normal state when $\rho(\lambda)$ diverges algebraically at $\lambda_{\text{max}}$. These Fermi and non-Fermi liquid normal states correspond to the low-energy states of class I and class II low-rank SYK models in Ref. [39].

Starting from these normal states, we examined the leading pairing instabilities in both spin-singlet and triplet channels by solving the linearized Schwinger-Dyson equations. The spin independent Yukawa interactions $g_{ij,k}(c_{i\uparrow}^{\dagger}c_{j\uparrow}\phi_k + c_{i\downarrow}^{\dagger}c_{j\downarrow}\phi_k)$, which model the charge fluctuations of correlated metals, show the pairing instabilities from both spin singlet and spin triplet channels. However, the spin dependent Yukawa interactions $g_{ij,k}(c_{i\uparrow}^{\dagger}c_{j\uparrow}\phi_k - c_{i\downarrow}^{\dagger}c_{j\downarrow}\phi_k)$, which represent the spin fluctuations, yield the leading pairing instabilities from the spin triplet channels $F^{1,2}(\tau) \sim \langle c_{\uparrow}^{\dagger}(\tau)c_{\uparrow}^{\dagger}(0) \pm c_{\downarrow}^{\dagger}(\tau)c_{\downarrow}^{\dagger}(0)\rangle$. Although both the spin-independent and dependent Yukawa interactions result in the same normal states, the resulting pairing instabilities are not the same. Furthermore, it is interesting to note that the critical temperature for the pairing state arising from the non-Fermi liquid is higher than that of the Fermi liquid (Figure 3). Although conventional wisdom may expect that the pairing would be eventually suppressed due to incoherence of the fermions, our theory demonstrates an example that the enhancement of the boson propagator, which glues the fermion pair, dominates the effect of the large fermion self-energy, which shortens each dressed fermion's lifetime. In this theory, there is no *ad hoc* parameter to control the relative contributions of the boson propagator and fermion self-energy to the pairing instabilities. The control knob of our theory $\rho(\lambda)$ influences both the enhancement of the pairing glue and the incoherence of the fermions, revealing a concrete physical meaning of the distribution $\rho(\lambda)$.

Since the Yukawa-SYK model is zero-dimensional, the natural follow-up question is the extension of our work to higher dimensions. If a quantum dot that consists of a large number of bosons and fermions realizes the paired state of the Yukawa-SYK model, we can consider an array of the coupled quantum dots as a higher dimensional generalization of our theory. Then, the leading spin-triplet pairing instabilities from the spin-dependent Yukawa interactions raise an interesting question: can the array of the coupled Yukawa-SYK quantum dots realize any unconventional (topological) superconductor? Furthermore, our analysis is based on the linearized Schwinger-Dyson equations. To examine the thermodynamic properties of the strongly interacting paired states below $T_c$, it would be interesting to explore the solutions of the full nonlinear Schwinger-Dyson equations.

# Acknowledgements

This work was supported by the NSERC of Canada and the Center for Quantum Materials at the University of Toronto. We would like to thank Xiangyu Cao for his friendly explanation about the low-rank SYK models.

# A Derivation of the Effective Action

We derive the effective action by averaging over the random Yukawa couplings, $g_{ij,k}$. Assuming that the model is self-averaging, we construct the large $N$ effective action from the disorder average of the partition function $\overline{Z}$ instead of the free energy $\overline{\log Z}$. In the language of the replica field theory, we are assuming that the replica diagonal terms dominate the low-energy

physics while the replica non-diagonal terms are suppressed by $\mathcal{O}(1/N)$.

$$
\begin{aligned}
e^{-S_\lambda} &= \overline{e^{-S_g}} \\
&= \prod_{k=1}^{M}\left[\prod_{i=1}^{N}\int \frac{dg_{ii,k}}{\sqrt{4\pi\lambda_k}}e^{-(g_{ii,k})^2/4\lambda_k-(g_{ii,k}/2N)\left(A_{ii,k}+A_{ii,k}^\dagger\right)}\right] \\
&\qquad\qquad \times\left[\prod_{i<j}\int \frac{dg_{ij,k}}{\sqrt{2\pi\lambda_k}}e^{-(g_{ij,k})^2/2\lambda_k-(g_{ij,k}/N)\left(A_{ij,k}+A_{ij,k}^\dagger\right)}\right] \\
&= \prod_{k=1}^{M}\left[\prod_{i=1}^{N}e^{\left(\lambda_k/4N^2\right)\left(A_{ii,k}+A_{ii,k}^\dagger\right)^2}\right]\left[\prod_{i\neq j}^{N}e^{\left(\lambda_k/4N^2\right)\left(A_{ij,k}+A_{ij,k}^\dagger\right)^2}\right] \\
&= \exp\left[\sum_{i,j=1}^{N}\sum_{k=1}^{M}\frac{\lambda_k}{4N^2}\left(A_{ij,k}+A_{ij,k}^\dagger\right)^2\right],
\end{aligned}
\tag{49}
$$

where $A_{ij,k}=\int_0^\beta d\tau\, c_{i\alpha}^\dagger\sigma_{\alpha\beta}^a c_{j\beta}\phi_k$. The summation is assumed for the repeated Greek indices. Therefore

$$
\begin{aligned}
S_\lambda &= -\sum_{i,j=1}^{N}\sum_{k=1}^{M}\int_0^\beta d\tau\,d\tau'\,\frac{\lambda_k}{2N^2}\phi_k(\tau)\phi_k(\tau')\sigma_{\alpha\beta}^a\sigma_{\alpha'\beta'}^a\Big[c_{i\alpha}^\dagger(\tau)c_{j\beta}(\tau)c_{j\alpha'}^\dagger(\tau')c_{i\beta'}(\tau') \\
&\qquad\qquad\qquad\qquad\qquad\qquad\qquad\qquad +c_{i\alpha}^\dagger(\tau)c_{j\beta}(\tau)c_{i\alpha'}^\dagger(\tau')c_{j\beta'}(\tau')\Big] \\
&= \frac{M}{2}\int_0^\beta d\tau\,d\tau'\left[\frac{1}{M}\sum_{k=1}^{M}\lambda_k\phi_k(\tau)\phi_k(\tau')\right] \\
&\quad \times\left\{\left[\frac{1}{N}\sum_{i=1}^{N}c_{i\alpha}^\dagger(\tau)c_{i\beta'}(\tau')\right]\sigma_{\alpha\beta}^a\left[\frac{1}{N}\sum_{j=1}^{N}c_{j\alpha'}^\dagger(\tau')c_{j\beta}(\tau)\right]\sigma_{\alpha'\beta'}^a \right. \\
&\qquad\qquad \left. -\left[\frac{1}{N}\sum_{i=1}^{N}c_{i\alpha}^\dagger(\tau)c_{i\alpha'}^\dagger(\tau')\right]\sigma_{\alpha\beta}^a\left[\frac{1}{N}\sum_{j=1}^{N}c_{j\beta'}(\tau')c_{j\beta}(\tau)\right]\sigma_{\alpha'\beta'}^a\right\} \\
&= \frac{M}{2}\int_0^\beta d\tau\,d\tau'D(\tau',\tau)\Big[G_{\beta'\alpha}(\tau',\tau)\sigma_{\alpha\beta}^a G_{\beta\alpha'}(\tau,\tau')\sigma_{\alpha'\beta'}^a \\
&\qquad\qquad\qquad\qquad -F_{\alpha'\alpha}^+(\tau',\tau)\sigma_{\alpha\beta}^a F_{\beta\beta'}(\tau,\tau')(\sigma^a)_{\beta'\alpha'}^T\Big] \\
&= \frac{M}{2}\int_0^\beta d\tau\,d\tau'D(\tau',\tau)\,\mathrm{tr}\Big[G(\tau',\tau)\sigma^a G(\tau,\tau')\sigma^a -F^+(\tau',\tau)\sigma^a F(\tau,\tau')(\sigma^a)^T\Big],
\end{aligned}
\tag{50}
$$

where "tr" is the trace over the spin indices. To impose the relationship between the bilocal fields and the fermions and bosons, we introduce the Lagrange multipliers:

$$
S_\Pi = \frac{1}{2}\int_0^\beta d\tau\,d\tau'\,\Pi(\tau,\tau')\left[MD(\tau',\tau)-\sum_{k=1}^{M}\lambda_k\phi_k(\tau)\phi_k(\tau')\right],
\tag{51}
$$

$$
S_\Sigma = -\int_0^\beta d\tau\,d\tau'\,\Sigma_{\alpha\alpha'}(\tau,\tau')\left[NG_{\alpha'\alpha}(\tau',\tau)-\sum_{i=1}^{N}c_{i\alpha}^\dagger(\tau)c_{i\alpha'}(\tau')\right],
\tag{52}
$$

$$
\begin{aligned}
S_\Phi = -\frac{1}{2}\int_0^\beta d\tau\,d\tau'\,\Phi_{\alpha\alpha'}(\tau,\tau')&\left[NF_{\alpha'\alpha}^+(\tau',\tau)-\sum_{i=1}^{N}c_{i\alpha}^\dagger(\tau)c_{i\alpha'}^\dagger(\tau')\right] \\
+\Phi_{\alpha\alpha'}^+(\tau,\tau')&\left[NF_{\alpha'\alpha}(\tau',\tau)-\sum_{i=1}^{N}c_{i\alpha}(\tau)c_{i\alpha'}(\tau')\right].
\end{aligned}
\tag{53}
$$

Let us define the Fourier transformations

$$c_{i\alpha}(\tau) = \frac{1}{\sqrt{\beta}} \sum_{n\in\mathbb{Z}} c_{i\alpha}(i\omega_n) e^{-i\omega_n \tau}, \tag{54}$$

$$\phi_k(\tau) = \frac{1}{\sqrt{\beta}} \sum_{n\in\mathbb{Z}} \phi_k(i\nu_n) e^{-i\nu_n \tau}, \tag{55}$$

where $\omega_n = (2n+1)\pi/\beta$ and $\nu_n = 2n\pi/\beta$ are the fermionic and bosonic Matsubara frequencies, respectively. Since the model is time-translation invariant, the bilocal fields are functions of $\tau - \tau'$. The consistent definition of the Fourier transformations for the bilocal fields is

$$G_{\alpha\alpha'}(\tau, \tau') = G_{\alpha\alpha'}(\tau - \tau') = \frac{1}{\beta} \sum_{n\in\mathbb{Z}} G_{\alpha\alpha'}(i\omega_n)^{-i\omega_n(\tau-\tau')}. \tag{56}$$

Then our modified action $\widetilde{S} = \widetilde{S}_c + \widetilde{S}_\phi + \widetilde{S}_\lambda$ including the Lagrange multipliers in the Fourier space is

$$\widetilde{S}_c = -\sum_{i=1}^{N} \sum_{n=0}^{\infty} f_i^\dagger(i\omega_n) \cdot \left[ \mathcal{G}_0(i\omega_n)^{-1} - \mathcal{S}(i\omega_n) \right] \cdot f_i(i\omega_n), \tag{57}$$

$$\widetilde{S}_\phi = \sum_{k=1}^{M} \sum_{n=1}^{\infty} \left( \nu_n^2/c^2 + m^2 - \lambda_k \Pi(i\nu_n) \right) |\phi_k(i\nu_n)|^2 + \frac{1}{2} \sum_{k=1}^{M} \left( m^2 - \lambda_k \Pi(0) \right) (\phi_k(0))^2 \tag{58}$$

$$\widetilde{S}_\lambda = -\frac{N}{2} \sum_{n\in\mathbb{Z}} \mathrm{Tr}\left[ \mathcal{S}(i\omega_n) \cdot \mathcal{G}(i\omega_n) \right], + \frac{M}{2} \sum_{n\in\mathbb{Z}} D(i\nu_n) \Big\{ \Pi(i\nu_n)$$

$$+ \frac{1}{\beta} \sum_{m\in\mathbb{Z}} \mathrm{tr}\left[ G(i\omega_m) \sigma^a G(i\omega_m + i\nu_n) \sigma^a \right] - \mathrm{tr}\left[ F^+(i\omega_m) \sigma^a F(i\omega_m + i\nu_n)(\sigma^a)^T \right] \Big\}, \tag{59}$$

where "Tr" is the trace over the indices for the four-component spinor

$$f_i(i\omega_n) = \begin{bmatrix} c_{i\uparrow}(i\omega_n) & c_{i\downarrow}(i\omega_n) & c_{i\uparrow}^\dagger(-i\omega_n) & c_{i\downarrow}^\dagger(-i\omega_n) \end{bmatrix}^T, \tag{60}$$

and

$$\mathcal{G}_0(i\omega_n)^{-1} = \begin{bmatrix} (i\omega_n + \mu)\sigma^0 & 0 \\ 0 & (i\omega_n - \mu)\sigma^0 \end{bmatrix}, \tag{61}$$

$$\mathcal{S}(i\omega_n) = \begin{bmatrix} \Sigma(i\omega_n) & \Phi(i\omega_n) \\ \Phi^+(i\omega_n) & -\Sigma(-i\omega_n)^T \end{bmatrix}, \tag{62}$$

$$\mathcal{G}(i\omega_n) = \begin{bmatrix} G(i\omega_n) & F(i\omega_n) \\ F^+(i\omega_n) & -G(-i\omega_n)^T \end{bmatrix}. \tag{63}$$

By integrating out the fermions and bosons, we obtain the effective action $S_{\mathrm{eff}} = S_0 + \widetilde{S}_\lambda$ in terms of the bilocal fields, where

$$S_0 = -N \sum_{n=0}^{\infty} \mathrm{Tr}\log\left[ \mathcal{G}_0(i\omega_n)^{-1} - \mathcal{S}(i\omega_n) \right] + \sum_{k=1}^{M} \sum_{n=1}^{\infty} \log\left( \nu_n^2/c^2 + m^2 - \lambda_k \Pi(i\nu_n) \right)$$

$$+ \sum_{k:\lambda_k < \lambda_{\max}} \frac{1}{2} \log\left( m^2 - \lambda_k \Pi(0) \right) + \frac{\beta N}{2} \left( m^2 - \lambda_{\max} \Pi(0) \right) \varphi. \tag{64}$$

$\varphi$ is the magnitude of the condensed bosons defined in Eq. (13).

When the set of the variances $\{\lambda_k\}$ form a well-defined distribution

$$\rho(\lambda) = \frac{1}{M}\sum_{k=1}^{M}\delta(\lambda - \lambda_k). \tag{65}$$

in the large $M$ limit, we can rewrite $S_{\text{eff}}$ as

$$
\begin{aligned}
S_{\text{eff}} = &-\frac{N}{2}\sum_{n\in\mathbb{Z}}\text{Tr}\log\left[\mathcal{G}_0(i\omega_n)^{-1} - \mathcal{S}(i\omega_n)\right]\\
&+\frac{M}{2}\sum_{n\neq 0}\int_0^{\lambda_{\max}}d\lambda\,\rho(\lambda)\log\left(\nu_n^2/c^2 + m^2 - \lambda\Pi(i\nu_n)\right) + \frac{\beta N}{2}\left(m^2 - \lambda_{\max}\Pi(0)\right)\varphi\\
&-\frac{N}{2}\sum_{n\in\mathbb{Z}}\text{Tr}\left[\mathcal{S}(i\omega_n)\cdot\mathcal{G}(i\omega_n)\right] + \frac{M}{2}\sum_{n\in\mathbb{Z}}D(i\nu_n)\Big\{\Pi(i\nu_n)\\
&+\frac{1}{\beta}\sum_{m\in\mathbb{Z}}\text{tr}\left[G(i\omega_m)\sigma^a G(i\omega_m + i\nu_n)\sigma^a\right] - \text{tr}\left[F^+(i\omega_m)\sigma^a F(i\omega_m + i\nu_n)(\sigma^a)^T\right]\Big\}. \tag{66}
\end{aligned}
$$

## B  Bose-Einstein Condensation for the $\eta > 0$ Model

In low-dimensional systems, the violent quantum fluctuations often prevent the presence of long-range order or the Bose-Einstein condensation (BEC). For example, in the Yukawa-SYK model with a fixed variance coupling, i.e., $\rho(\lambda) \sim \delta(\lambda - \lambda_0)$, bosons are not condensed although the strong Yukawa interactions with fermions renormalize their mass to zero [28, 36]. However, the low dimensionality does not always rule out the possibility of BEC. In this appendix, we demonstrate that our model with $\eta > 0$ (class I) can show the Bose-Einstein condensation despite strong quantum fluctuations due to the zero-dimensional nature of the all-to-all interactions.

The Bose-Einstein condensation occurs when the number of excited states is bounded from above, i.e., $N_{\text{excited}} < N_0$, at some temperatures below $T_{\text{BEC}}$. If there were $N$ number of bosons, the remaining $N - N_0$ number of bosons are forced to be in the ground state since the Bose statistics limits the maximum number of the bosons in the excited states. Then, the macroscopic number ($N - N_0 \gg 1$) of bosons are said to be condensed in the ground state.

For our model, it is difficult to calculate the maximum number of available excited states $N_0$ explicitly. Instead, we can demonstrate that the bosons inevitably become unstable (i.e., $m^2 - \lambda_{\max}\Pi(0) < 0$) without the Bose-Einstein condensation when $\eta > 0$. If there were no BEC ($\varphi = 0$), we will first show that the boson Green function $D(i\nu_n)$ is bounded from above when $\eta > 0$. Next, we will prove that the fermion self-energy $\Sigma(i\omega_n)$ is bounded from above because $D(i\nu_n)$ is bounded from above. At last, when the fermion self-energy is bounded from above, we can show that the boson self-energy $\Pi(0)$ at zero frequency is bounded from below. The lower bound of $\Pi(0)$ is a function of temperature, and we will show that $m^2 - \lambda_{\max}\Pi(0)$ inevitably becomes negative at some finite temperatures because of this lower bound. In short, we will prove that BEC must occur in the $\eta > 0$ model because "no BEC" implies the bounded boson Green function which results in the unstable bosons. Below, we mathematically demonstrate this idea.

When $\eta > 0$, we first demonstrate below that the contribution of uncondensed bosons to the boson Green function $D_N(i\nu_n)$ is bounded from above. From Eq. (26), we find

$$D_N(i\nu_n) \le D^{(0)}(0)f_\eta(1) = \frac{\lambda_{\max}}{m^2}f_\eta(1) = \frac{\lambda_{\max}}{\eta m^2}. \tag{67}$$

Using l'Hospital's rule, explicit calculations can show that $\lim_{x \to 1} f_\eta(x) = 1/\eta$. Since $f_\eta$ is a strictly increasing function of $x \in (-\infty, 1]$, $f_\eta(x) \leq f_\eta(1) = 1/\eta$. Thus, the above inequality holds. Note that if $\eta \leq 0$, $f_\eta(x)$ diverges at $x = 1$.

Let us suppose that the bosons are not condensed, i.e., $\varphi = 0$. Then, from Eq. (18),

$$
\begin{aligned}
|\beta i \Sigma_0(i\omega_n)|^2 &= \left| \sum_{n' \in \mathbb{Z}} \gamma D(i\omega_{n'} - i\omega_n) i G_0(i\omega_{n'}) \right|^2 \\
&\leq \sum_{n' \in \mathbb{Z}} |\gamma D(i\omega_{n'} - i\omega_n) i G_0(i\omega_{n'})|^2 = \sum_{n' \in \mathbb{Z}} |\gamma D(i\omega_{n'} - i\omega_n)|^2 |i G_0(i\omega_{n'})|^2 \\
&\leq \left( \frac{\gamma \lambda_{\max}}{\eta m^2} \right)^2 \sum_{n' \in \mathbb{Z}} |i G_0(i\omega_{n'})|^2 \equiv \mathcal{S}^2 \,.
\end{aligned}
\tag{68}
$$

Note that the normal state fermion Green function without the Bose-Einstein condensation has the canonical form:

$$
G_0(i\omega_n) = \frac{1}{i\omega_n - \Sigma_0(i\omega_n)} \,.
\tag{69}
$$

Since $|i G_0(i\omega_n)|^2$ decays at least as fast as $1/n^2$, its sum over all Matsubara frequencies must be convergent. Hence, $\mathcal{S}$ must be a finite nonnegative real number. To be more specific, if $|\Sigma_0(i\omega_n)| \gg |\omega_n|$ as $|\omega_n| \to \infty$, $\sum_n |i G_0(i\omega_n)|^2$ converges because the square of the Green function decays faster than $1/n^2$. If $|\Sigma_0(i\omega_n)| \ll |\omega_n|$ as $|\omega_n| \to \infty$, then $\sum_n |i G_0(i\omega_n)|^2$ is also convergent because $|i G_0(i\omega_n)|^2$ decays as $1/n^2$.

When the fermion self-energy is bounded from above, the boson self-energy at zero frequency is bounded from below:

$$
\begin{aligned}
\Pi(0) &= \frac{2}{\beta} \sum_{n \in \mathbb{Z}} (i G_0(i\omega_n))^2 = \frac{2}{\beta} \sum_{n \in \mathbb{Z}} \frac{1}{[\omega_n + i\Sigma_0(i\omega_n)]^2} = \sum_{n \in \mathbb{Z}} \frac{2\beta}{[(2n+1)\pi + \beta i \Sigma_0(i\omega_n)]^2} \\
&\geq \sum_{n=0}^{\infty} \frac{4\beta}{[(2n+1)\pi + |\beta i \Sigma_0(i\omega_n)|]^2} \\
&\geq \sum_{n=0}^{\infty} \frac{4\beta}{[(2n+1)\pi + \mathcal{S}]^2} = \frac{\beta}{\pi^2} \psi^{(1)} \left( \frac{1}{2} + \frac{\mathcal{S}}{2\pi} \right),
\end{aligned}
\tag{70}
$$

where $\psi^{(1)}$ is the polygamma function of order 1. Thus, $m^2 - \lambda_{\max} \Pi(0) < 0$ if

$$
T < \lambda_{\max} \psi^{(1)}(1/2 + \mathcal{S}/2\pi)/\pi^2 m^2 \,.
\tag{71}
$$

Since $\mathcal{S}$ also depends on temperature $T$, one can question whether the above inequality can be satisfied for some finite temperatures $T > 0$. To examine the existence of the solution of the inequality, let us first investigate the $\gamma = 0$ case for given $\eta$, $m$, and $\lambda_{\max}$. Since $\mathcal{S} = 0$ when $\gamma = 0$,

$$
\Pi(0) > (\beta/\pi^2)\psi^{(1)}(1/2) = (\beta/\pi^2)(\pi^2/2) = \beta/2 \,.
\tag{72}
$$

Thus, $m^2 - \lambda_{\max} \Pi(0) < 0$ if $T < \lambda_{\max}/2m^2$, i.e., the bosons become unstable at finite temperatures if there were no Bose-Einstein condensation. Suppose $i\Sigma_0(i\omega_n)$ for finite $\gamma > 0$ is continuously connected to the $\gamma = 0$ limit, i.e., if we increase $\gamma$ from 0, then $\mathcal{S}$ also continuously increases from zero. By numerically solving the Schwinger-Dyson equations, we confirmed that the self-consistent Green functions and self-energies for finite $\gamma > 0$ is continuously connected to the self-consistent solution for $\gamma = 0$ (up to $\gamma = 1$. This is also previously confirmed

both numerical calculations and analytical perturbation theory in [39]. Since $\psi^{(1)}(1/2+\mathcal{S}/2\pi)$ is an analytic function of $\mathcal{S} \geq 0$, the existence of the solution at $\gamma = 0$ implies the existence of the solution for some finite $\gamma > 0$.

To sum up, the Bose-Einstein condensation is necessary for the $\eta > 0$ model to avoid the instability of bosons. If the bosons are condensed ($\varphi \neq 0$), the bosons can be either critical or stable because the total boson Green function $D(i\nu_n) = (\beta\lambda_{\max}/\gamma)\varphi\delta_{n,0} + D_N(i\nu_n)$ is no longer bounded from above. Then the quantum fluctuations of the bosons can yield arbitrarily large fermion self-energy. When the fermion self-energy is sufficiently large, it will not result in large boson self-energy which makes bosons unstable. Note that our conclusion is consistent with the previous work for the fixed-variance model. Since the boson Green function is not bounded from above for the fixed-variance model, the bosons can remain critical without the condensation [28]. We would also like to note that the physics of our model is different from that of free Bose gas at $d = 0$ because the bosons are strongly coupled to the massless fermions. The intertwined dynamics of the strongly correlated bosons and fermions can result in the Bose condensation even at $d = 0$ under certain conditions.

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
