# Peer review of "Pairing Instabilities of the Yukawa-SYK Models with Controlled Fermion Incoherence"

_SciPost Physics, doi:SciPost Phys. 12, 151 (2022)_

## Round 1 · Referee Report · Anonymous (Referee 1) · 2021-11-4

Report

This is an interesting paper. The authors generalize the Yukawa-SYK model to allow for a continuous distribution of variances to the bosonic fields. This allows them to tune the normal state from Fermi to non-Fermi behavior, and to study how this influences the critical temperature of the low-T superconducting state, and its nature (singlet or triplet).

The main conclusion is that Tc may be enhanced for the non-Fermi liquid because even though the quasiparticles become less coherent, the same "knob" strengthens the coupling to bosons, and the latter wins.

I think this is a nice demonstration that this phenomenology is possible, in a model where an essentially exact solution can be found after self-averaging.

Nevertheless, I do not see how this work meets any of the acceptance criteria (i) Detail a groundbreaking theoretical/experimental/computational discovery; (ii) Present a breakthrough on a previously-identified and long-standing research stumbling block; (iii) Open a new pathway in an existing or a new research direction, with clear potential for multipronged follow-up work; (iv) Provide a novel and synergetic link between different research areas.

This is the reason I suggest the paper be moved/accepted to SciPost Physics Core, whose criteria it meets better.

One optional suggestion for the authors: perhaps if you allow the masses of the bosons to also follow a distribution (instead of all being set to m) you might get an independent "knob" on the behavior of the bosons, and be able to tune the non-Fermi character of the fermions independently of the strength of the boson glue?

  • validity: top
  • significance: good
  • originality: good
  • clarity: high
  • formatting: excellent
  • grammar: good

Author:  Wonjune Choi  on 2022-03-18  [id 2300]

(in reply to Report 1 on 2021-11-04)
Category:
objection

We first thank the anonymous referee for reviewing our manuscript. Although the referee questioned the qualification of our paper to be published in SciPost Physics, we respectfully disagree.

Anonymous referee writes:

I think this is a nice demonstration that this phenomenology is possible, in a model where an essentially exact solution can be found after self-averaging.

Nevertheless, I do not see how this work meets any of the acceptance criteria (i) Detail a groundbreaking theoretical/experimental/computational discovery; (ii) Present a breakthrough on a previously-identified and long-standing research stumbling block; (iii) Open a new pathway in an existing or a new research direction, with clear potential for multipronged follow-up work; (iv) Provide a novel and synergetic link between different research areas.

This is the reason I suggest the paper be moved/accepted to SciPost Physics Core, whose criteria it meets better.

Our response:

The recurring theme in the field of the correlated superconductivity is understanding the competition between the non-Fermi liquid behaviour and superconducting instability due to strong electronic interactions. Despite many theoretical studies have been done since the discovery of high $T_c$ cuprates, the answer to the question is still inconclusive because the controlled theories of strongly correlated systems are rare.

Our work (ii) presents a breakthrough on a previously-identified and long-standing research stumbling block by investigating the solvable models which allow the systematic theoretical studies on the competition between the non-Fermi liquid behaviour and the pairing instability. The Yukawa-SYK model with the distribution of variances is a rare exactly solvable model which supports both the Fermi liquid, non-Fermi liquids, and paired states at low temperatures. This can be contrasted to the previously published variants of the SYK model. They support only the non-Fermi liquid states and the paired states. We would also like to highlight that our model does not introduce any ad hoc parameter to independently control the amount of fermion incoherence and pairing instability. The distribution of variances, which has a concrete physical meaning, impacts on both fermion pairing and fermion incoherence. Thus, our work examined the competition between non-Fermi liquidness and superconductivity by controlling the common physical origin.

By examining the pairing instability of both Fermi and non-Fermi liquid normal states, we could draw non-trivial but reliable conclusion that the non-Fermi liquid normal states with more incoherent fermions can have stronger tendency to form Cooper pairs (i.e., higher transition temperature $T_c$) compared to the Fermi liquid. Given that our model is exactly solvable, our theoretical conclusion can motivate follow-up researches for more realistic and complex models for the superconductivity born out of non-Fermi liquid normal states.

Furthermore, our work expand the scope of the SYK model researches by examining the pairing instabilities of the non-fast-scrambling normal states of the SYK model in the systematic fashion. While most previous researches focused on the singlet pairing of the fast scrambling NFL states of the SYK model, our work called for attention to both spin singlet and triplet pairings of non-fast-scrambling normal states of the Yukawa-SYK models.

With the one parameter family of the Fermi and non-Fermi liquid states controlled by the parameter $\eta$, we discovered the leading spin triplet pairing instability born out of various amount of fermion incoherence. As our research first explored the leading triplet pairing instabilities from a single Yukawa-SYK quantum dot, there are plethora of follow-up questions such as possibility of topological spin triplet SYK superconductivity from some higher dimensional generalisation of our theory. Therefore, our work (iii) opened a new pathway in the field of the SYK superconductivity.

Anonymous referee writes:

One optional suggestion for the authors: perhaps if you allow the masses of the bosons to also follow a distribution (instead of all being set to m) you might get an independent "knob" on the behavior of the bosons, and be able to tune the non-Fermi character of the fermions independently of the strength of the boson glue?

Our response:

We thank the referee for the optional suggestion to expand our research. Introducing the distribution of mass must be an interesting research direction to investigate the role of boson mass on the pairing instability and non-Fermi liquid behaviour of the normal states. However, in the current work, the introduction of the variance of the coupling constant was sufficient to demonstrate how the fermion incoherence competes with the Cooper pair formation. Therefore, we will leave the extensive survey of our model with the distribution of the boson mass to the future work.

---

## Round 1 · Referee Report · Yuxuan Wang (Referee 2) · 2021-12-5

Strengths

Exactly solvability of the model which addresses an important and challenging theoretical problem.

Weaknesses

The discussion on the normal-state properties is a bit too terse; reference to a previous work should be accompanied by a more detailed discussion to make the paper more self-contained.

Report

In this interesting work, the authors generalized the Yukawa-SYK model (equivalent to the low-rank SYK model) to allow for more structure in the random coupling constant (g_{ijk} is a Gaussian random variable in ij, but obeys a more complicated distribution in k). As a result, the authors obtain a rich phase diagram displaying Fermi-liquid, non-Fermi liquid, and superconducting phases. This work is an important contribution to the understanding of the myriad of phases of a strongly-correlated system which usually defies analytical approaches. For this reason I think it meets the criteria for publication in SciPost.

I do have several questions which I hope they can clarify:

1. It would be helpful to compare the non-Fermi liquid obtained here with that of the Yukawa-SYK model with a fixed-variance coupling. Do the Green's functions have similar scaling behavior? Is there a limiting case of $\eta$ for which the results go back to that with a fixed variance? Naively, one would think if \eta \to -\infinity, the only possible value of \lambda is \lambda_{max}. It would also be nice if the authors can compare the properties of superconducting transition/state with those obtained in Ref. 36 and in Phys. Rev. B 104, 125120 (2021).

2. The author mentioned that for some parameter range the boson field can condense, leading to a constant contribution to the boson propagator. However, it has been argued in Ref. 28 (among several other works cited by the authors) the boson cannot condense even if its renormalized mass vanishes, essentially due to strong quantum fluctuations in 0d, but rather remains critical. I understand the authors are considering a different model, but it would be helpful if the author can provide more explanations on this point.

  • validity: top
  • significance: high
  • originality: good
  • clarity: high
  • formatting: perfect
  • grammar: perfect

Author:  Wonjune Choi  on 2022-03-18  [id 2299]

(in reply to Report 2 by Yuxuan Wang on 2021-12-05)
Category:
answer to question

We thank Professor Yuxuan Wang for his sincere review and for considering our work suitable for publication in SciPost Physics.

Prof. Wang writes:

The discussion on the normal-state properties is a bit too terse; reference to a previous work should be accompanied by a more detailed discussion to make the paper more self-contained.

Our response:

Following the suggestion, we expand Section 4, which discusses the normal state properties of the Yukawa-SYK model, to make our paper more self-contained. Below, we answer the questions in the referee's report.

Prof. Wang writes:

  1. It would be helpful to compare the non-Fermi liquid obtained here with that of the Yukawa-SYK model with a fixed-variance coupling. Do the Green's functions have similar scaling behavior?

Our response:

The Yukawa-SYK models with a fixed variance coupling [Esterlis & Schmalian (2019) and Classen & Chubukov (2021)] demonstrate two distinct nontrivial normal states at finite temperatures: the quantum critical SYK non-Fermi liquid (SYK-NFL) normal state at low temperatures, and the impurity-like non-Fermi liquid state at intermediate temperature window.

The SYK-NFL state is the fast scrambling, conformal solution of the Yukawa-SYK model. The non-Fermi liquid nature originates from strong boson-fermion interactions which dynamically tune the mass of bosons to zero [Wang (2020)]. The scaling property of the fermion Green function $G(i\omega_n)$ is dominated by the fermion self-energy $\Sigma(i\omega_n) \sim i\mathrm{sgn}(\omega_n) |\omega_n|^{1-2\Delta}$ at low frequencies. The scaling dimension lies between $\frac{1}{4} < \Delta < \frac{1}{2}$ and depends on the ratio between the number of bosons ($M$) and fermions ($N$), $\gamma = M / N$.

The normal states of our model with the distribution of the variances $\rho(\lambda)$ show different scaling behaviour compared to the SYK-NFL states. Both "Fermi liquid" ($\eta > 0$) and "non-Fermi liquid" ($-1 < \eta < 0$) states show the "impurity-like" behaviour, which is also observed from the fixed-variance model at intermediate temperature window [Esterlis & Schmalian (2019) and Classen & Chubukov (2021)]. The strong interactions do not tune the mass of the boson to zero. Instead, the bosons act as impurity centres that scatter fermions. Especially, as we can see from Eq. (23), the impurity-like scattering due to the Bose condensate $\varphi$ and the static uncondensed bosons $D_N(i\nu_n = 0)$ (the zero frequency part of the uncondensed boson's Green function) lead to the leading scaling dimension of the fermions $\Delta = \frac{1}{2}$, i.e., $G(i\omega_n) \sim i\mathrm{sgn}(\omega_n)$, for all $\gamma$ and $\eta > -1$. The subleading self-energy corrections $\Sigma_N(i\omega_n) \sim i\mathrm{sgn}(\omega_n) |\omega_n|^{1+\eta}$ due to the impurity-like scattering from the uncondensed bosons introduce anomalously large quasiparticle decay $1/\tau_{\mathrm{life}} \sim T^{1+\eta}$ [Kim, Cao & Altman (2020a)].

While the impurity-like non-Fermi liquid fixed point is not stable in the fixed-variance models, our model supports the impurity-like Fermi liquid and non-Fermi liquid states as stable infrared fixed points at low temperatures. When it comes to the scaling property of the stable IR fixed point, our model with the distribution of the variances shows qualitatively distinct normal states compared to the previously studied fixed-variance models which support the SYK-NFL state. Furthermore, by controlling the distribution $\rho(\lambda)$, our model provides systematic control over the fermion incoherence.

Prof. Wang writes:

Is there a limiting case of $\eta$ for which the results go back to that with a fixed variance? Naively, one would think if \eta \to -\infinity, the only possible value of \lambda is \lambda_{max}.

Our response:

Since the model distribution $\rho \sim (\lambda_{\max} - \lambda)^\eta$ in Eq. (16) is a well-defined probability distribution only if $\eta > -1$, i.e., $\int_0^{\lambda_{\max}} \rho(\lambda) ~ d\lambda$ diverges if $\eta \leq -1$, we cannot naively extend our result to the limit $\eta \to -\infty$. If we tried to extend the calculations for $\eta \leq -1$, the integral in Eq. (24) for the boson Green function diverges.

Although no limiting case of $\eta$ can fully reproduce the results of the model with a fixed variance, $\eta \to -1^{+}$ limit partially recovers the impurity-like NFL behaviour and its pairing instability in Esterlis & Schmalian (2019). The model distribution $\rho(\lambda)$ in Eq. (16) is designed to explore diverse singular properties of the boson Green function $D_N(i\nu_n)$ near zero frequency, $\nu_0 = 0$. The singular property of $D_N(i\nu_n)$ is determined by the function $f_{\eta}(x)$, whose asymptotic property is derived in Eq. (28). For $\eta > -1$, we can examine the boson Green function such that $f_{\eta}(x)$ decays slower than $\frac{1}{1-x}$. To be more specific, our model distribution $\rho_\eta(\lambda)$ parameterized by $\eta > -1$ allowed us to explore the boson Green function

$$ D_{\eta}(i\nu_n) \sim D^{(0)}(i\nu_n)(1- D^{(0)}(i\nu_n)\Pi(i\nu_n))^\eta $$

for $\nu_n$ near 0. Note that if we consider a fixed variance model, i.e., $\rho(\lambda) \sim \delta(\lambda-\lambda_{\max})$, then

$$ D_{\mathrm{fixed}}(i\nu_n) \sim D^{(0)}(i\nu_n) (1- D^{(0)}(i\nu_n) \Pi(i\nu_n))^{-1}. $$

We can see that the low-frequency behaviour of the fixed variance model $D_{\mathrm{fixed}}$ is equal to that of $D_\eta$ with $\eta \to -1^{+}$. (For unambiguous notation, we denote $D_\eta$ for the boson Green function with the distribution of variances and $D_{\mathrm{fixed}}$ for the boson Green function with a fixed variance.)

In addition, the asymptotic behaviour of the fermion self-energy of our model $\Sigma_{\eta}$ approaches the self-energy of the fixed variance model $\Sigma_{\mathrm{impurity}}$ for the impurity-like NFL state in Esterlis & Schmalian (2019):

$$\Sigma(i\omega_n) \sim i \mathrm{sgn}(\omega_n) |\omega_n|^{1+\eta} \to \Sigma_{\mathrm{impurity}}(i\omega_n) \sim i \mathrm{sgn}(\omega_n) $$

as $\eta \to -1^{+}$.

Therefore, the impurity-like NFL state of the fixed variance coupling model can be considered as the limiting case of "$\eta \to -1^{+}$".

However, we should note that the stability of the low-energy fixed point is dramatically changed: while the impurity-like fixed point is stable for $\eta > -1$, it is an unstable fixed point in case of the fixed variance model (see Figure 2 of Esterlis & Schmalian (2019)). The SYK-NFL state becomes the stable IR fixed point in the fixed variance model.

Prof. Wang writes:

It would also be nice if the authors can compare the properties of superconducting transition/state with those obtained in Ref. 36 and in Phys. Rev. B 104, 125120 (2021).

Our response:

Both Ref. 36 and Phys. Rev. B 104, 125120 (2021) discussed the pairing instabilities of the impurity-like NFL state in the strongly coupled regime, i.e., $\rho(\lambda) \sim \delta(\lambda-\lambda_{\max})$ with $\lambda_{\max} \gg m^3$. For the spin-singlet Yukawa coupling ($S_g \sim g_{ij,k} c_{i\alpha}^\dagger c_{j\alpha} \phi_k$ in Eq. (4)), the nature of the paired state and the pairing transition is qualitatively similar to the pairing instability of the impurity-like NFL state discussed in Ref. 36 and Phys. Rev. B 104, 125120 (2021). However, in case of the spin-triplet Yukawa coupling ($S_g \sim g_{ij,k} c_{i\alpha}^\dagger \sigma^z_{\alpha\beta} c_{j\beta} \phi_k$), we found that the spin-triplet pairing $F^{1,2}(\tau) \sim c_{i\uparrow}^\dagger(\tau) c_{i\uparrow}^\dagger(0) \pm c_{i\downarrow}^\dagger(\tau) c_{i\downarrow}^\dagger(0)$ at the distant time $\tau >0$ is the leading instability.

Most importantly, our work focused on how the transition temperature changes as we tune the fermion incoherence by controlling the distribution $\rho(\lambda)$. In the fixed variance model, changing the absolute magnitude of the variance $\lambda_{\max}$ or the mass of bosons $m$ does not influence the critical temperature $T_c$. This can be seen from the constant $T_c$ in Figure 5 of Esterlis & Schmalian (2019). Previous work demonstrated one particular example of how the impurity-NFL state experiences pairing instability. However, in our theory, the fermion incoherence is strongly dependent on the distribution control parameter $\eta$. Hence, we could demonstrate how the fermion incoherence born out of the strong fermion-boson interactions competes with the enhanced pairing tendency born out of the same physical origin. This could not be discussed in the previous studies due to the lack of controllability for the fermion incoherence.

Prof. Wang writes:

  1. The author mentioned that for some parameter range the boson field can condense, leading to a constant contribution to the boson propagator. However, it has been argued in Ref. 28 (among several other works cited by the authors) the boson cannot condense even if its renormalized mass vanishes, essentially due to strong quantum fluctuations in 0d, but rather remains critical. I understand the authors are considering a different model, but it would be helpful if the author can provide more explanations on this point.

The Bose-Einstein condensation (BEC) occurs when the number of excited states is bounded from above, i.e., $N_{\mathrm{excited}} < N_0$, at some temperatures below $T_{\mathrm{BEC}}$. If there were $N$ number of bosons, the remaining $N - N_0$ number of bosons are forced to be in the ground state since the Bose statistics limits the maximum number of the excited bosons. Then, the macroscopic number ($N - N_0 \gg 1$) of bosons are said to be condensed in the ground state.

For our model, it is difficult to calculate the maximum number of available excited states $N_0$ explicitly. Instead, we can demonstrate that the bosons inevitably become unstable (i.e., $m^2 - \lambda_{\max}\Pi(0) < 0$) without the Bose-Einstein condensation when $\eta > 0$. If there were no BEC ($\varphi = 0$), we will first show that the boson Green function $D(i\nu_n)$ is bounded from above when $\eta > 0$. Next, we will prove that the fermion self-energy $\Sigma(i\omega_n)$ is bounded from above because $D(i\nu_n)$ is bounded from above. At last, when the fermion self-energy is bounded from above, we can show that the boson self-energy $\Pi(0)$ at zero frequency is bounded from below. The lower bound of $\Pi(0)$ is a function of temperature, and we will show that $m^2 - \lambda_{\max} \Pi(0)$ inevitably becomes negative at some finite temperatures because of this lower bound. In short, we will prove that BEC must occur in $\eta>0$ model because "no BEC" implies the bounded boson Green function which results in the unstable bosons. Below, we mathematically demonstrate this idea.

When $\eta > 0$, we first demonstrate below that the contribution of uncondensed bosons to the boson Green function $D_N(i\nu_n)$ is bounded from above. From Eq. (25), we find

$$D_N(i\nu_n) \leq D^{(0)}(0) f_{\eta}(1)= \frac{\lambda_{\max}}{m^2}f_{\eta}(1) = \frac{\lambda_{\max}}{\eta m^2}.$$

Using l'Hospital's rule, explicit calculations can show that $\lim_{x\to1} f_\eta(x) = 1/\eta.$ Since $f_\eta$ is a strictly increasing function of $x \in (-\infty, 1]$, $f_\eta(x) \leq f_\eta(1)=1/\eta$. Thus, the above inequality holds. Note that if $\eta \leq 0$, $f_\eta(x)$ diverges at $x=1$.

Let us suppose that the bosons are not condensed, i.e., $\varphi = 0$. Then, from Eq. (18),

$$ \begin{align} |\beta i\Sigma_0(i\omega_n)|^2 &= \left|\sum_{n'\in\mathbb{Z}} \gamma D(i\omega_{n'} - i\omega_n) iG_0(i\omega_{n'}) \right|^2 \ &\leq \sum_{n' \in \mathbb{Z}} \left| \gamma D(i\omega_{n'} - i\omega_n) iG_0(i\omega_{n'}) \right|^2 \ &= \sum_{n' \in \mathbb{Z}} \left| \gamma D(i\omega_{n'} - i\omega_n) \right|^2 \left| iG_0(i\omega_{n'}) \right|^2 \ &\leq \left( \frac{\gamma\lambda_{\max}}{\eta m^2}\right)^2 \sum_{n'\in\mathbb{Z}} | iG_0(i\omega_{n'})|^2 \equiv \mathcal{S}^2 \end{align} $$

Note that the normal state fermion Green function without the Bose-Einstein condensation has the canonical form:

$$G_0(i\omega_n) = \frac{1}{i\omega_n - \Sigma_0(i\omega_n)}.$$

Since $|iG_0(i\omega_n)|^2$ decays at least as fast as $1/n^2$, its sum over all Matsubara frequencies must be convergent. Hence, $\mathcal{S}$ must be a finite nonnegative real number. To be more specific, if $|\Sigma_0(i\omega_n)| \gg |\omega_n|$ as $|\omega_n| \to \infty$, $\sum_{n} |i G_0(i\omega_n)|^2$ converges because the square of the Green function decays faster than $1/n^2$. If $|\Sigma_0(i\omega_n)| \ll |\omega_n|$ as $|\omega_n| \to \infty$, then $\sum_{n} |i G_0(i\omega_n)|^2$ is also convergent because $|iG_0(i\omega_n)|^2$ decays as $1/n^2$.

When the fermion self-energy is bounded from above, the boson self-energy at zero frequency is bounded from below:

$$ \begin{align} \Pi(0) &= \frac{2}{\beta} \sum_{n\in\mathbb{Z}} (iG_0(i\omega_n))^2 = \frac{2}{\beta}\sum_{n\in\mathbb{Z}} \frac{1}{[\omega_n + i\Sigma_0(i\omega_n)]^2} \ &= \sum_{n\in\mathbb{Z}} \frac{2\beta}{[(2n+1)\pi + \beta i\Sigma_0(i\omega_n)]^2} \ &\geq \sum_{n=0}^{\infty} \frac{4\beta}{[(2n+1)\pi + |\beta i\Sigma_0(i\omega_n)|]^2} \ &\geq \sum_{n=0}^{\infty} \frac{4\beta}{[(2n+1)\pi + \mathcal{S}]^2} = \frac{\beta}{\pi^2}\psi^{(1)}\left(\frac{1}{2}+\frac{\mathcal{S}}{2\pi}\right), \end{align} $$

where $\psi^{(1)}$ is the polygamma function of order 1. Thus, $m^2 - \lambda_{\max}\Pi(0) < 0$ if

$$T < \lambda_{\max} \psi^{(1)}(1/2+\mathcal{S}/2\pi) / \pi^2 m^2.$$

Since $\mathcal{S}$ also depends on temperature $T$, one can question whether the above inequality can be satisfied for some finite temperatures $T>0$. To examine the existence of the solution of the inequality, let us first investigate the $\gamma = 0$ case for given $\eta$, $m$, and $\lambda_{\max}$. Since $\mathcal{S} = 0$ when $\gamma = 0$,

$$\Pi(0) > (\beta / \pi^2) \psi^{(1)}(1/2) = (\beta / \pi^2)(\pi^2/2) = \beta/2.$$

Thus, $m^2 - \lambda_{\max} \Pi(0) < 0$ if $T < \lambda_{\max} / 2m^2$, i.e., the bosons become unstable at finite temperatures if there were no Bose-Einstein condensation. Suppose $i\Sigma_0(i\omega_n)$ for finite $\gamma > 0$ is continuously connected to the $\gamma = 0$ limit, i.e., if we increase $\gamma$ from $0$, then $\mathcal{S}$ also continuously increases from zero. (By numerically solving Schwinger-Dyson equations, we confirmed that the self-consistent Green functions and self-energies for finite $\gamma > 0$ is continuously connected to the self-consistent solution for $\gamma = 0$ (up to $\gamma = 1$). This is also previously confirmed by Kim, Cao & Altman (2020b).) Since $\psi^{(1)}(1/2+\mathcal{S}/2\pi)$ is an analytic function of $\mathcal{S} \geq 0$, the existence of the solution at $\gamma = 0$ implies the existence of the solution for some finite $\gamma > 0$.

In conclusion, the Bose-Einstein condensation is necessary for the $\eta > 0$ model to avoid the instability of bosons. If the bosons are condensed ($\varphi \neq 0$), the bosons can be either critical or stable because the total boson Green function $D(i\nu_n) = (\beta \lambda_{\max}/\gamma) \varphi \delta_{n,0} + D_N(i\nu_n)$ is no longer bounded from above. Then the quantum fluctuations of the bosons can yield arbitrarily large fermion self-energy. The sufficiently large fermion self-energy would not result in too large boson self-energy which can make the bosons unstable. Note that our conclusion is consistent with the previous work for the fixed-variance model. Since the boson Green function is not bounded from above for the fixed-variance model, the bosons can remain critical without the condensation [Wang (2020)]. We would also like to note that physics of our model is different from that of free Bose gas at $d=0$ because the bosons are strongly coupled to the massless fermions. The intertwined dynamics of the strongly correlated bosons and fermions can result in the Bose condensation even at $d=0$ under certain conditions.

---

## Round 2 · Author Response

Dear Editor,

Thank you for arranging the review of our article. We are grateful to all referees for carefully reading our paper.
We largely expanded Section 4 about the normal state properties of the Yukawa-SYK models following the suggestion of one of the reviewers, Prof. Yuxuan Wang. Also, we wrote another appendix to explain why the Bose-Einstein condensation can occur in class I systems despite strong quantum fluctuations due to low dimensionality. We further highlighted how our work is distinguished from the previous works; we investigated a one-parameter family of states, whose fermion incoherence can be systematically controlled rather than being fixed.

Sincerely,

Wonjune Choi
Omid Tavakol
Yong Baek Kim

---

## Round 2 · List of Changes

1. Major revision on Section 4 with detailed discussions about the normal state properties of the Yukawa-SYK models
2. New appendix about the Bose-Einstein condensation in class I systems.
3. Include more discussions to compare and contrast our work from previous research works on the SYK model and its variants.

---

## Editorial Decision

published